# *GraphVid*: It Only Takes a Few Nodes to Understand a Video

**Abstract.** We propose a concise representation of videos that encode perceptually meaningful features into graphs. With this representation, we aim to leverage the large amount of redundancies in videos and save computations. First, we construct superpixel-based graph representations of videos by considering superpixels as graph nodes and create spatial and temporal connections between adjacent superpixels. Then, we leverage Graph Convolutional Networks to process this representation and predict the desired output. As a result, we are able to train models with much fewer parameters, which translates into short training periods and a reduction in computation resource requirements. A comprehensive experimental study on the publicly available datasets Kinetics-400 and Charades shows that the proposed method is highly cost-effective and uses limited commodity hardware during training and inference. **It reduces the computational requirements 10-fold** while achieving results that are comparable to state-of-the-art methods. We believe that the proposed approach is a promising direction that could open the door to solving video understanding more efficiently and enable more resource limited users to thrive in this research field.

## 1 Introduction

The field of video understanding has gained prominence thanks to the rising popularity of videos, which has become the most common form of data on the web. On each new uploaded video, a variety of tasks can be performed, such as tagging [18], human action recognition [38], anomaly detection [47], etc. New video-processing algorithms are continuously being developed to automatically organize the web through the flawless accomplishment of the aforementioned tasks.

Nowadays, Deep Neural Networks are the de-facto standard for video understanding [36]. However, with every addition of a new element to the training set (that is, a full training video), more resources are required in order to satisfy the enormous computational needs. On the one hand, the exponential increment in the amount of data raises concerns regarding our ability to handle it in the future. On the other hand, it has also spurred an highly creative research field aimed at finding ways to mitigate this burden.

Among the first-generation of video processing methods were ones geared toward adopting 2D convolution neural networks (CNNs), due to their computational efficiency [44]. Others decomposed 3D convolutions [14, 57] into simpler operators, or split a complex neural network into an ensemble of lightweight networks [9]. However, video understanding has greatly evolved since then, with the current state-of-the-art methods featuring costly attention mechanisms [4, 20, 32, 3, 15, 6, 30]. Beyond accuracy, a

prominent advantage of the latest generation of methods is that they process raw data, that is, video frames that do not undergo any advanced pre-processing. Meanwhile, pursuing new video representations and incorporating pre-computed features to accelerate training is a promising direction that requires more extensive research.

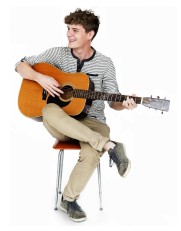
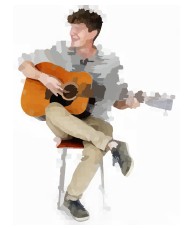

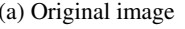

(a) Original image          (b) Mean superpixels

Fig. 1: A visual comparison between a pixel and a mean-superpixel representation. On the left, the original image is presented. On the right, we present the image formed by generating superpixel regions using SLIC and filling each region with its mean color.

Prior to the renaissance of deep learning [29], much research was done on visual feature generation. Two prominent visual feature generation methods are superpixels[1] and optic-flow[2]. These techniques' ability to encode perceptually meaningful features has greatly contributed to the success of computer vision algorithms. Superpixels provide a convenient, compact representation of images that can be very useful for computationally demanding problems, while optic-flow provides hints about motion. We rely on these methods to construct a novel representation of videos that encodes sufficient information for video understanding: 1) adjacent pixels are grouped together in the form of superpixels, and 2) temporal relations and proximities are expressed via graph connectivity. The example depicted in Figure 1 provides an intuition for the sufficiency of superpixel representation for scene understanding. It contains the superpixel regions obtained via SLIC [2], with each region filled with the mean color. One can clearly discern a person playing a guitar in both images. A different way of depicting the relations between superpixels is a graph with nodes representing superpixels [34, 11, 5]. Such a representation has the advantage of being invariant to rotations and flips, which obviates the need for further augmentation. We here demonstrate how this representation can reduce the computational requirements for processing videos.

Recent years have seen a surge in the utilization of Graph Neural Networks (GNNs) [26] in tasks that involve images [34, 11, 5], audio [12, 62] and other data forms [55, 56, 1]. In this paper, we propose *GraphVid*, a concise graph representation of videos that enables video processing via GNNs. *GraphVid* constructs a graph representation of videos

---

[1] Superpixel techniques segment an image into regions by considering similarity measures, defined using perceptual features.

[2] Optic-flow is the pattern of the apparent motion of an object(s) in the image between two consecutive frames due to the movement of the object or the camera.

that is subsequently processed via a GCN to predict a target. We intend to exploit the power of graphs for efficient video processing. To the best of our knowledge, we are the first to utilize a graph-based representation of videos for efficiency. *GraphVid* dramatically reduces the memory footprint of a model, enabling large batch-sizes that translate to better generalization. Moreover, it utilizes models with an order-of-magnitude fewer parameters than the current state-of-the-art models while preserving the predictive power. **In summary, our contributions are:**

1. We present *GraphVid* - a simple and intuitive, yet sufficient representation of video clips. This simplicity is crucial for delivering efficiency.
2. We propose a dedicated GNN for processing the proposed representation. The proposed architecture is compared with conventional GNN models in order to demonstrate the importance of each component of *GraphVid*.
3. We present 4 types of new augmentations that are directly applied to the video-graph representation. A thorough ablation study of their configurations is preformed in order to demonstrate the contribution of each.
4. We perform a thorough experimental study, and show that *GraphVid* greatly outperforms previous methods in terms of efficiency - first and foremost, the paper utilizes GNNs for efficient video understanding. We show that it successfully reduces computations while preserving much of the performance of state-of-the-art approaches that utilize computationally demanding models.

## 2   Related Work

### 2.1   Deep Learning for Video Understanding

CNNs have found numerous applications in video processing [33, 50, 60]. These include LSTM-based networks that perform per-frame encoding [45, 51, 60] and the extension of 2D convolutions to the temporal dimension, *e.g.*, 3D CNNs such as C3D [49], R2D [44] and R(2+1)D [50].

The success of the Transformer model [52] has led to the development of attention-based models for vision tasks, via self-attention modules that were used to model spatial dependencies in images. NLNet [54] was the first to employ self-attention in a CNN. With this novel attention mechanism, NLNet possible to model long-range dependencies between pixels. The next model to be developed was GCNet [7], which simplified the NL-module, thanks to its need for fewer parameters and computations, while preserving its performance. A more prominent transition from CNNs to Transformers began with Vision Transformer (ViT) [13], which prompted research aimed at improving its effectiveness on small datasets, such as Deit [48]. Later, vision-transformers were adapted for video tasks [35, 4, 6, 15, 30, 32], now crowned as the current state-of-the-art that top the leader-boards of this field.

The usage of graph representation in video understanding sparsely took place in the work of Wang [55]. They used a pre-trained Resnet variants [22] on the MSCOCO dataset [31] in order to generate object bounding boxes of interest on each video frame. These bounding boxes are later used for the construction of a spatio-temporal graph that describes how objects change through time, and perform classification on top of

the spatio-temporal graph with graph convolutional neural networks [26]. However, we note that the usage of a large backbone for generating object bounding boxes is harmful for performance. We intend to alleviate this by proposing a lighter graph representation. In combination of a dedicated GNN architecture, our representation greatly outperforms [55] in all metrics.

## 2.2    Superpixel Representation of Visual Data

Superpixels are groups of perceptually similar pixels that can be used to create visually meaningful entities while heavily reducing the number of primitives for subsequent processing steps [46]. The efficiency of the obtained representation has led to the development of many superpixel-generation algorithms for images [46]. This approach was adapted for volumetric data via the construction of supervoxels [37], which are the trivial extension to depth. These methods were adjusted for use in videos [58] by treating the temporal dimension as depth. However, this results in degraded performance, as inherent assumptions regarding neighboring points in the 3D space do not apply to videos with non-negligible motion. Recent approaches especially designed to deal with videos consider the temporal dimensions for generating superpixels that are coherent in time. Xu *et a.l* [59] proposed a hierarchical graph-based segmentation method. This was followed by the work of Chang *et a.l* [8], who suggested that Temporal Superpixels (TSPs) can serve as a representation of videos using temporal superpixels by modeling the flow between frames with a bilateral Gaussian process.

## 2.3    Graph Convolutional Neural Networks

Introduced in [26], Graph Convolutional Networks (GCNs) have been widely adopted for graph-related tasks [61, 28]. The basic form of a GCN uses aggregators, such as average and summation, to obtain a node representation given its neighbors. This basic form was rapidly extended to more complex architectures with more sophisticated aggregators. For instance, Graph Attention Networks [53] use dot-product-based attention to calculate weights for edges. Relational GCNs [42] add to this framework by also considering multiple edge types, namely, relations (such as temporal and spatial relations), and the aggregating information from each relation via separate weights in a single layer. Recently, GCNs have been adopted for tasks involving audio [12, 62] and images [34, 11, 5]. Following the success of graph models to efficiently perform image-based tasks, we are eager to demonstrate our extension of the image-graph representation to videos.

## 3    *GraphVid* - A Video-Graph Representation

In this section, we introduce the methodology of *GraphVid*. First, we present our method for video-graph representation generation, depicted in Figure 2 and described in Algorithm 1. Then, we present our training methodology that utilizes this representation. Finally, we discuss the benefits of *GraphVid* and propose several augmentations.

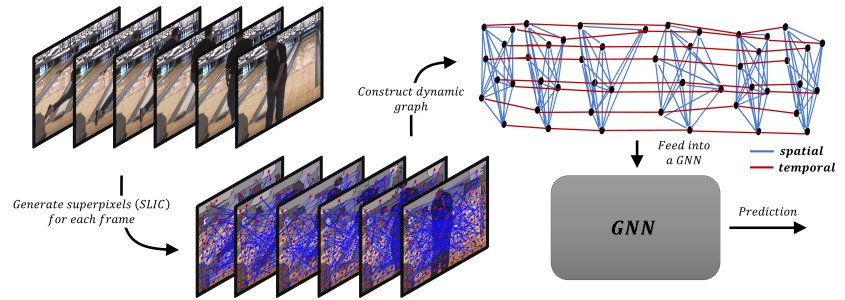

Fig. 2: The flow of *GraphVid*. Given a video clip, we generate superpixels using SLIC for each frame. The superpixels are used to construct a region-adjacency graph of a frame, with superpixels as nodes. Then, the graph sequence is connected via temporal proximities to construct a dynamic graph, which is later fed into a GNN for prediction.

### 3.1 Overview

In our framework, we deal with video clips that are sequences of $T$ video frames $v \in \mathbb{R}^{T \times C \times H \times W}$. The goal is to transform $v$ into a graph that is sufficiently informative for further processing. To achieve this, we use SLIC [2] to generate $S$ segmented regions, called *superpixels*, over each frame. We denote each segmented region as $R_{t,i}$, where $t \in [T]$ represents the temporal frame index, and $i \in [S]$ the superpixel-segmented region index. The following is a description of how we utilize the superpixels to construct our video-graph representation.

*Graph Elements* - We define the undirected graph $\mathcal{G}$ as a 3-tuple $\mathcal{G} = (\mathcal{V}, \mathcal{E}, \mathcal{R})$, where $\mathcal{V} = \{R_{t,i} | t \in [T], i \in [S]\}$ is the set of nodes representing the segmented regions, $\mathcal{E}$ is the set of labeled edges (to be defined hereunder) and $\mathcal{R} = \{spatial, temporal\}$ is a set of relations as defined in [42]. Each node $R_{t,i}$ is associated with an attribute $R_{t,i}.c \in \mathbb{R}^3$ representing the mean RGB color in that segmented region. Additionally, we refer to $R_{t,i}.y$ and $R_{t,i}.x$ as the coordinates of the superpixel's centroid, which we use to compute the distances between superpixels. These distances, which will later serve as the edge attributes of the graph, are computed by

$$d_{i,j}^{t_q \to t_p} = \sqrt{\left(\frac{R_{t_q,i}.y - R_{t_p,j}.y}{H}\right)^2 + \left(\frac{R_{t_q,i}.x - R_{t_p,j}.x}{W}\right)^2}. \quad (1)$$

Here, $t_q, t_p \in [T]$ denote frame indices, and $i, j \in [S]$ denote superpixel indices generated for the corresponding frames. The set of edges $\mathcal{E}$ is composed of two types: **1)** intra-frame edges (denoted $\mathcal{E}^{spatial}$) - edges between nodes corresponding to superpixels in the same frame. We refer to these edges as *spatial edges*. **2)** inter-frame edges (denoted $\mathcal{E}^{temporal}$) - edges between nodes corresponding to superpixels in two sequential frames. We refer to these edges as *temporal edges*. Finally, the full set of edges is given by $\mathcal{E} = \mathcal{E}^{spatial} \cup \mathcal{E}^{temporal}$. Following is a description of how we construct both components.

*Spatial Edges* - In similar to [5], we generate a region-adjacency graph for each frame, with edge attributes describing the distances between superpixel centroids. The notation $\mathcal{E}_t^{spatial}$ refers to the set of the spatial-edges connecting nodes corresponding to superpixels in the frame $t$, and $\mathcal{E}^{spatial} = \bigcup_{t=1}^{T} \mathcal{E}_t^{spatial}$. Each edge $e_{i,j}^t \in \mathcal{E}^{spatial}$ is associated with an attribute that describes the euclidean distance between the two superpixel centroids $i$ and $j$ in frame $t$, that is, $d_{i,j}^{t \to t}$. These distances provide information about the relations between the superpixels. Additionally, the distances are invariant to rotations and image-flips, which eliminates the need for those augmentations. Note that normalization of the superpixels' centroid coordinates is required in order to obscure information regarding the resolution of frames, which is irrelevant for many tasks, such as action classification. In Figure 3, we demonstrate the procedure of spatial edge generation for a cropped image that results in a partial graph of the whole image. Each superpixel is associated with a node, which is connected via edges to other adjacent nodes (with the distances between the superpixels' centroids serving as edge attributes).

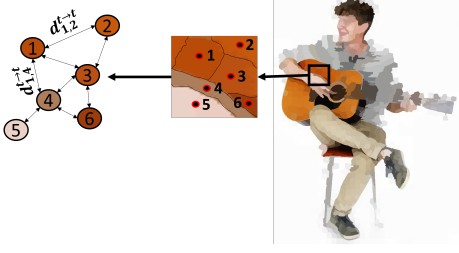

Fig. 3: Spatial edge generation. First, superpixels are generated. Each superpixel is represented as a node, which is connected via its edges to other such nodes within a frame. Each node is assigned the mean color of the respective segmented region, and each edge is assigned the distances between the superpixel centroids connected by that edge.

*Temporal Edges* - In modeling the temporal relations, we aim to connect nodes that tend to describe the same objects in subsequent frames. To do so, we rely on the assumption that in subsequent frames, such superpixels are attributed similar colors and the same spatial proximity. To achieve this, for each superpixel $R_{t,i}$, we construct a neighborhood $\mathcal{N}_{t,i}$ that contains superpixels from its subsequent frame whose centroids have a proximity of at most $d_{proximity} \in (0, 1]$ with respect to the euclidean distance. Then, we find the superpixel with the most similar color in this neighborhood. As a result, the $t^{th}$ frame is associated with the set of edges $\mathcal{E}_{t \to t+1}^{temporal}$ that model temporal relations with its subsequent frame, formally:

$$\mathcal{N}_{t,i} = \{R_{t+1,j} | d_{i,j}^{t \to t+1} < d_{proximity}\}, \tag{2}$$

$$neighbor(R_{t,i}) = \underset{R_{t+1,j} \in \mathcal{N}_{t,i}}{\operatorname{argmin}} |R_{t,i}.c - R_{t+1,j}.c|_2, \tag{3}$$

$$\mathcal{E}_{t \to t+1}^{temporal} = \{(R_{t,i}, temporal, neighbor(R_{t,i}) | i \in [S]\}. \tag{4}$$

Equipped with these definitions, we define the set of temporal edges connecting nodes corresponding to superpixels in frame $t$ to superpixels in frame $t + 1$ as the union of the temporal edge sets generated for all the frames: $\mathcal{E}^{temporal} = \bigcup_{t=1}^{T-1} \mathcal{E}_{t \to t+1}^{temporal}$ .

---

**Algorithm 1** Graph Generation

---

**Input:** $v \in \mathbb{R}^{T \times C \times H \times W}$                                          ▷ The input video clip
**Parameters:** $S \in \mathbb{N}$                              ▷ Number of superpixels per frame
                    $d_{proximity} \in (0, 1]$                        ▷ Diameter of neighborhoods
**Output:** $\mathcal{G} = (\mathcal{V}, \mathcal{E}, \mathcal{R})$                                          ▷ A video-graph
$\mathcal{V}, \mathcal{V}_{last}, \mathcal{E}^{spatial}, \mathcal{E}^{temporal} \leftarrow \emptyset, \emptyset, \emptyset, \emptyset$
**for** $t \in [T]$ **do**
      $SP \leftarrow SLIC(v[t], S)$
      $\mathcal{V} \leftarrow \mathcal{V} \cup SP$
      $\mathcal{E}^{spatial} \leftarrow \mathcal{E}^{spatial} \cup regionAdjacetEdges(SP)$
      $\mathcal{E}_{t-1 \to t}^{temporal} \leftarrow \emptyset$
      **for** $R_{t-1,i} \in \mathcal{V}_{last}$ **do**
            $\mathcal{N}_{t-1,i} \leftarrow \{R_{t,j} | d_{i,j}^{t-1 \to t} < d_{proximity}\}$
            $nn_{t-1,i} \leftarrow \operatorname{argmin}_{R_{t,j} \in \mathcal{N}_{t,i}} |R_{t,i}.c - R_{t,j}.c|_2)$
            $\mathcal{E}_{t-1 \to t}^{temporal} \leftarrow \mathcal{E}_{t-1 \to t}^{temporal} \cup \{(R_{t-1,i}, temporal, nn_{t-1,i})\}$
      **end for**
      $\mathcal{E}^{temporal} \leftarrow \mathcal{E}^{temporal} \cup \mathcal{E}_{t-1 \to t}^{temporal}$
      $\mathcal{V}_{last} \leftarrow SP$
**end for**
**return** $\mathcal{G} = (\mathcal{V}, \mathcal{E} = \mathcal{E}^{spatial} \cup \mathcal{E}^{temporal}, \mathcal{R} = \{spatial, tempo\})$

---

### 3.2 Model Architecture

In order to model both the spatial and temporal relations between superpixels, our model primarily relies on the Neural Relational Model [42], which is an extension of GCNs [26] to large-scale relational data. In a Neural Relational Model, the propagation model for calculating the forward-pass update of a node, denoted by $v_i$, is defined as

$$h_i^{(l+1)} = \sigma \left( \sum_{r \in \mathcal{R}} \sum_{j \in \mathcal{N}_i^r} \frac{1}{c_{i,r}} W_r^{(l)} h_j^{(l)} + W_0^{(l)} h_i^{(l)} \right), \qquad (5)$$

where $\mathcal{N}_i^r$ denotes the set of neighbor indices of node $i$ under relation $r \in \mathcal{R}$ (not to be confused with the notation $\mathcal{N}_{t,i}$ from Eq. 2). $c_{i,r}$ is a problem-specific normalization constant that can either be learned or chosen in advance (such as $c_{i,r} = |\mathcal{N}_i^r|$). To incorporate edge features, we adapt the approach proposed in [10], that concatenates node and edge attributes as a layer's input, yielding the following:

$$h_i^{(l+1)} = \sigma \left( \sum_{r \in \mathcal{R}} \sum_{j \in \mathcal{N}_i^r} \frac{1}{c_{i,r}} W_r^{(l)} [h_j^{(l)}, e_{i,j}] + W_0^{(l)} h_i^{(l)} \right), \qquad (6)$$

where $e_{i,j}$ is the feature of the edge connecting nodes $v_i, v_j$.

### 3.3   Augmentations

We introduce a few possible augmentations that we found useful for training our model as they improved the generalization.

*Additive Gaussian Edge Noise (AGEN)* -  Edge attributes represent distances between superpixel centroids. The coordinates of those centroids may vary due to different superpixel shapes with different centers of mass. To compensate for this, we add a certain amount of noise to each edge attribute. Given a hyper-parameter $\sigma_{edge}$, for each edge attribute $e_{u,v}$ and for each training iteration, we sample a normally distributed variable $z_{u,v} \sim N(0, \sigma_{edge})$ that is added to the edge attribute.

*Additive Gaussian Node Noise (AGNN)* -  Node attributes represent the colors of regions in each frame. Similar to edge attributes, the mean color of each segmented region may vary due to different superpixel shapes. To compensate for this, we add a certain amount of noise to each node attribute. Given a hyper-parameter $\sigma_{node}$, for each node attribute $v.c$ of dimension $d_c$ and for each training iteration, we sample a normally distributed variable $z_v \sim N_{d_c}(0, \sigma_{node} \cdot I_{d_c})$ that is added to the node attribute.

*Random Removal of Spatial Edges (RRSE)* -  This augmentation tends to mimic the regularization effect introduced in DropEdge [40]. Moreover, since the removal of edges leads to fewer message-passings in a GCN, this also accelerates the training and inference. To perform this, we choose a probability $p_{edge} \in [0, 1]$. Then, each edge $e$ is preserved with a probability of $p_{edge}$.

*Random Removal of Superpixels (RRS)* -  SLIC [2] is sensitive to its initialization. Consequently, each video clip may have several graph representations during different training iterations and inference. This can be mitigated by removing a certain amount of superpixels. The outcome is fewer nodes in the corresponding representative graph, as well as fewer edges. Similar to RRE, we choose a probability $p_{node} \in [0, 1]$ so that each superpixel is preserved with a probability of $p_{node}$.

### 3.4   Benefits of *GraphVid*

*Invariance Qualification* -  The absence of coordinates leads to invariance in the spatial dimension of each frame. It is evident that such a representation is invariant to rotation, horizontal flip and vertical flip, since the relations between different parts of the image are solely characterized by distances. This, in turn, obviates the need to perform such augmentations during training.

*Efficiency* -  We argue that our graph-based representation is more efficient than raw frames. To illustrate this, let $T, C, H$ and $W$ be the original dimensions of the video clip; that is, the number of frames, number of channels in each frame and height and width of a frame, respectively. This implies that the raw representation requires $T \cdot C \cdot H \cdot W$ parameters to encode a single input. Now, to calculate the size of the graph-video representation, let $S$ be the number of superpixels in a single frame. By construction, there

are at most $4 \cdot S$ edges in each frame because SLIC constraints each superpixel to have 4 adjacent superpixels. Each edge contains 3 values, corresponding to the distance on the image grid, source node and target node. Additionally, there are, at most, $S$ edges between every temporal step. This results in $3 \cdot ( \underbrace{4 \cdot S}_{\substack{intra \\ frame \\ edges}} + \underbrace{(T-1) \cdot S}_{\substack{inter \\ frame \\ edges}} ) + C \cdot \underbrace{T \cdot S}_{\substack{super- \\ pixels}}$ pa-

rameters in total. Typically, the second representation requires much fewer parameters because we choose $S$ so that $S \ll H \cdot W$.

*Prior Knowledge Incorporation* - Optical-flow and over-segmentation are encoded within the graph-video representation using the inter-frame and intra-frame edges. This incorporates strong prior knowledge within the resultant representation. For example, optical-flow dramatically improved the accuracy in the two-stream methodology that was proposed in [44]. Additionally, over-segmentation using superpixels has been found useful as input features for machine learning models due to the limited loss of important details, accompanied by a dramatic reduction in the expended time by means of reducing the number of elements of the input [21, 11, 5].

## 4   Experiments

We validated *GraphVid* on 2 human-action-classification benchmarks. The goal of human action classification is to determine the human-involved action that occurs within a video. The objectives of this empirical study were twofold:

– Analyze the impact of the various parameters on the accuracy of the model.
– As we first and foremost target efficiency, we sought to examine the resources' consumption of *GraphVid* in terms of Floating Point Operations (FLOPs). We followed the conventional protocol [16], which uses single-clip FLOPs as a basic unit of computational cost. We show that we are able to achieve a significant improvement in efficiency over previous methods while preserving state-of-the-art performance.

### 4.1   Setup

*Datasets* - We utilize two commonly used datasets for action classification: *Kinetics-400 (K400)* [23] and *Charades* [43]. Kinetics-400 [23] is a large-scale video dataset released in 2017 that contains 400 classes, with each category consisting of more than 400 videos. It originally had, in total, around 240K, 19K, and 38K videos for training, validation and testing subsets, respectively. Kinetics datasets are gradually shrinking over time due to videos being taken offline, making it difficult to compare against less recent works. We used a dataset containing 208K, 17K and 33K videos for training, validation and test respectively. We report on the most recently available videos. Each video lasts approximately 10 seconds and is assigned a label. The Charades dataset [43] is composed of 9,848 videos of daily indoor activities, each of an average length of 30 seconds. In total, the dataset contains 66,500 temporal annotations for 157 action classes. In the standard split, there are 7,986 training videos and 1,863 validation videos, sampled at 12 frames per second. We follow prior arts by reporting the Top-1 and Top-5 recognition accuracy for Kinetics-400 and mean average precision (mAP) for Charades.

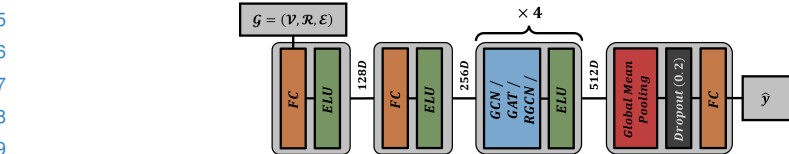

Fig. 4: The general graph neural network architecture we use throughout our experimental study.

*Network Architecture and Training* -  We use GNN variants as backbones for our experiments and feed each of them with our video-graph representation. Specifically, we consider Graph Convolutional Networks [26] (denoted GCNs), Graph Attention Networks [53] (denoted GATs) and Relational Graph Convolutional Networks [42] (denoted RGCNs). The general architecture of our backbones is depicted in Figure 4. It consists of 2 fully-connected (FC) layers with exponential linear unit (ELU) activations that transform the node feature vectors into a $256D$ feature space. Then come $4$ layers of the corresponding GNN layer type (that is, either GCN, GAT or RGCN along with an edge feature concatenation from Eq. 6) with a hidden size of 512 with ELU activations, followed by global mean pooling, dropout with a probability of $0.2$ and a linear layer whose output is the predicted logits. For the GAT layers, we use 4 attention heads in each layer, and average the attention heads' results to obtain the desired hidden layer size. For the RGCN layers, we specify 2 relations, which correspond to the spatial and temporal relations, as described in Section 3. We use the Adam [25] with a learning rate of $1e-3$ for optimization and do not change it throughout the training.

We divide the videos into clips using a sliding window of 20 frames, using a stride of 2 between every 2 consecutive frames and a stride of 10 between every 2 consecutive video clips. In all the experiments, we used a fixed batch size of 200, which captures the context of a time window that endures $200 \times 20 = 4000$ frames per batch.

*Inference* -  At the test phase, we use the same sliding window methodology as in the training. We follow the common practice of processing multiple views of a long video and average per-view logits to obtain the final results. The number of views is determined by the validation dataset.

*Implementation Details* -  All the experiments were run on a Ubuntu 18.04 machine with Intel i9-10920X, 93GB RAM and 2 GeForce RTX 3090 GPUs. Our implementation of *GraphVid* is in Python3. Specifically, to generate superpixels, we use the SLIC [2] algorithm via its implementation *fast-slic* [24]. To generate graphs and train the graph neural models, we use Pytorch-Geometric [19]. We use a fixed seed for SLIC's initialization and cache the generated graphs during the first training epochs in order to further reduce the number of computations.

## 4.2   Ablation Study

We conduct an in-depth study on Kinetics-400 to analyze the performance gain contributed by incorporating the different components of *GraphVid*.

*Graph Neural Network Variants and Number of Superpixels per Frame -* We assess the performance of different GNN variants: GCN [26] is trained without edge relations (*i.e.* temporal and spatial edges are treated via the same weights). GAT [53] is trained by employing the attention mechanism for neighborhood aggregation without edge relations. RGCN [42] is trained with edge relations, as described in Section 3.2.

The results of the action classification on Kinetics-400 are shown in Figure 5. In this series, the number of views is fixed at 8, which is the number of views that was found to be most effective for the validation set. For all variants, increasing the number of superpixels per frame ($S$) contributes to the accuracy of the model. We notice a significant improvement in accuracy for the lower range of the number of superpixels, while the accuracy begins to saturate for $S \geq 650$. Increasing further the number of superpixels leads to bigger inputs, which require more computations. As our goal is to maximize the efficiency, we do not experiment with larger inputs in this section. We

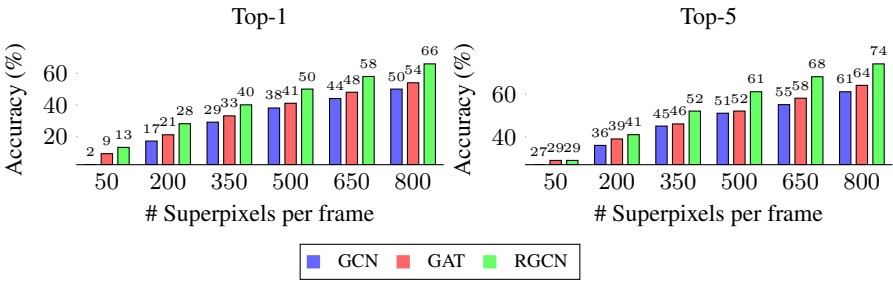

Fig. 5: The effect of varying the number of superpixels per frame on test accuracy on Kinetics-400.

further present in Table 1 the models' specifications for 800 superpixels, which is the best-performing configuration in this series of experiments. Not surprisingly, the simple GCN variant requires the least amount of computations among the three. Meanwhile, the RGCN variant requires fewer computations than GAT and achieves a higher level of accuracy. We conclude that it is beneficial to incorporate edge relations when wishing to encode temporal and spatial relations in videos, and that those features are not easily learned by heavy computational models, such as GAT.

Table 1: Comparison of model specifications for various architectures. We report the Top-1 and Top-5 accuracy on Kinetics-400.

| Model | Top-1 | Top-5 | FLOPs ($\cdot 10^9$) | Params ($\cdot 10^6$) |
|-------|-------|-------|------|--------|
| $GCN$ | 50.1 | 61.6 | 28 | 2.08 |
| $GAT$ | 54.7 | 64.5 | 56 | 3.93 |
| $RGCN$ | 66.2 | 74.1 | 42 | 2.99 |

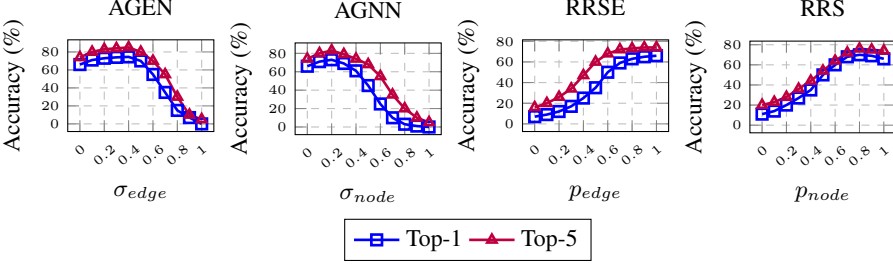

Fig. 6: The impact of the proposed augmentations on test accuracy of Kinetics-400: Additive Gaussian edge noise (AGEN). Additive Gaussian node noise (AGNN). Random removal of spatial edges (RRSE). Random removal of superpixels (RRS).

*Augmentations* - We assessed the impact of augmentations on the test's performance and their ability to alleviate over-fitting. For this purpose, we chose the best configuration obtained from the previous experiments, that is, RGCN with 800 superpixels per frame, and trained it while adding one type of augmentation at a time. The results of this series are depicted in Figure 7. Each graph shows the level of accuracy reached by training the model with one of the various parameters that control the augmentation.

We begin with the analysis of the AGEN and AGNN. Both augmentations relate to the addition of Gaussian noise to the attributes of the edges and the nodes of the input graphs, with the corresponding parameters controlling the standard deviation of that Gaussian noise. The impact of these augmentations is less noticeable as these parameters head towards 0, since lower values reflect the scenarios in which little or no augmentations are performed. Continuously increasing the parameter slightly brings about a gradual improvement in the accuracy, until a turning point is reached, after which the level of accuracy starts to decline until it reaches $\sim \frac{1}{400}$, which resembles a random classifier. The decrease in accuracy stems from the noise obscuring the original signal, allegedly forcing the classifier to classify noise that is not generalizable to the test set. In the cases of RRSE and RRS, the random removal of spatial edges harms the accuracy of the model for all values of $p_{edge} < 1$. This finding leads us to conclude that spatial edges encode meaningful information about spatial relations between the superpixel entities. Moreover, slightly removing the nodes positively impacts the level of accuracy, reaching a peak at around $p_{node} \approx 0.8$. To conclude this series, we present the values that lead to the best Top-1 accuracy score in Table 2.

Table 2: Augmentation parameters and their optimized values.

| Param | $\sigma_{edge}$ | $\sigma_{node}$ | $p_{edge}$ | $p_{node}$ |
|-------|-------|-------|-------|-------|
| Value | 0.4 | 0.2 | 1 | 0.8 |
| Top-1 | 74.5 | 73 | 66 | 70 |
| Top-5 | 85 | 83 | 74 | 76 |

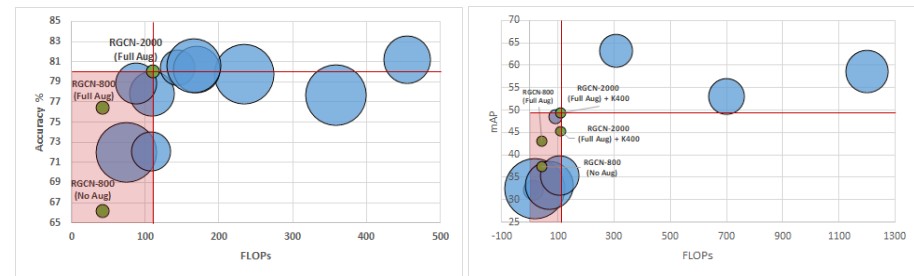

(a) FLOPS vs Kinetics-400 Accuracy        (b) FLOPS vs Charades mAP

Fig. 7: **Model FLOPs vs. performance -** Green bubbles indicates *GraphVid* variants from Table 3 and Table 4. Identities of the other models are omitted in order to avoid overload on the plot. *GraphVid* achieves comparable performance to the state-of-the-art while greatly reducing the number of parameters and FLOPs. For Kinetics-400, RGCN-2000 with the full set of augmentations achieves almost the same performance as all computationally heavy models on the plot, while requiring the least amount of parameters and FLOPs. For Charades, RGCN-2000 with the full set of augmentations and pretraining on Kinetics-400 is on par with the state-of-the-art, and fewer compute requirements. Bubble radius indicates the number of parameters of the model.

### 4.3   Comparison to the State-of-the-Art

*Kinetics-400 -* We present the Kinetics-400 results for our RGCN model variant in Table 3 and Figure 7a, along with comparisons to previous arts, including convolutional-based and transformer-based methods. Our results are denoted RGCN-$d$, where $d$ represents the number of superpixels. Additionally, we use the set of augmentations with the individually optimized hyper-parameters from Table 2 to train these models. First, when the RGCN-800 model is trained with the full set of augmentations (denoted Full-Aug), it achieves a significantly higher Top-1 accuracy than when it is trained without any augmentation (denoted No-Aug) or when each augmentation is applied individually. These results demonstrate the effectiveness of our model and that our carefully designed augmentations can alleviate overfitting and improve the generalization over the test set. Second, all our RGCNs require orders-of-magnitude fewer computations than the current state-of-the-art architectures, as well as more than $\times 10$ fewer parameters.

*Charades -* We train 2 RGCN variants with 800 and 2000 superpixels per frame with the same set of augmentations and hyper-parameters found in Table 2. Additionally, we follow prior arts [17, 15] by pre-training on K-400 followed by replacing the last FC layer to match the output dimensionality and fine-tuning on Charades. Table 4 and Figure 7b show that when our RGCN model is trained with 2000 superpixels per frame, its mAP score is comparable to the current state-of-the-art, but this score is reached with orders-of-magnitude fewer computations and using considerably fewer parameters than prior arts.

Table 3: Comparisons to state-of-the-art on the Kinetics-400 dataset. We report the Top-1 and Top-5 accuracy scores. The top section of the table depicts convolution-based models. The middle section depicts transformer-based models, and the bottom section represents our graph-based models.

| Method | Top-1 | Top-5 | Views | FLOPs ($\cdot 10^9$) | Param ($\cdot 10^6$) |
|---|---|---|---|---|---|
| SlowFast R101+N [17] | 79.8 | 93.9 | 30 | 234 | 59.9 |
| X3D-XXL R101+N [16] | 80.4 | 94.6 | 30 | 144 | 20.3 |
| MViT-B, 32×3 [15] | 80.2 | 94.4 | 5 | 170 | 36.6 |
| TimeSformer-L [6] | 80.7 | 94.7 | 3 | 2380 | 121.4 |
| ViT-B-VTN [35] | 78.6 | 93.7 | 1 | 4218 | 11.04 |
| ViViT-L/16x2 [4] | 80.6 | 94.7 | 12 | 1446 | 310.8 |
| Swin-S [32] | 80.6 | 94.5 | 12 | 166 | 49.8 |
| Swin-B [32] | 82.7 | 95.5 | 12 | 282 | 88.1 |
| RGCN-800 (No Aug) | 66.2 | 74.1 | 8 | **42** | **2.57** |
| RGCN-800 (Full Aug) | 76.4 | 91.1 | 8 | **42** | **2.57** |
| RGCN-2000 (Full Aug) | 80.0 | 94.3 | 8 | **110** | **2.57** |

Table 4: Comparisons to state-of-the-art on the Charades multi-label dataset. We report the mAP scores as more than one ground truth action is possible.

| Method | mAP | FLOPs ($\cdot 10^9$) | Params ($\cdot 10^6$) |
|---|---|---|---|
| MoVieNet-A2 [27] | 32.5 | 6.59 | 4.8 |
| MoVieNet-A4 [27] | 48.5 | 90.4 | 4.9 |
| MoVieNet-A6 [27] | 63.2 | 306 | 31.4 |
| TVN-1 [39] | 32.2 | 13 | 11.1 |
| TVN-4 [39] | 35.4 | 106 | 44.2 |
| AssembleNet-50 [41] | 53.0 | 700 | 37.3 |
| AssembleNet-101 [41] | 58.6 | 1200 | 53.3 |
| SlowFast 16 × 8 R101 [17] | 45.2 | 7020 | 59.9 |
| RGCN-800 (No Aug) | 37.4 | **42** | **2.57** |
| RGCN-800 (Full Aug) | 43.1 | **42** | **2.57** |
| RGCN-2000 (Full Aug) | 45.3 | **110** | **2.57** |
| RGCN-2000 (Full Aug)+K400 | 49.4 | **110** | **2.57** |

## 5   Conclusions and Future Work

In this paper, we present *GraphVid*, a graph video representations that enable video-processing via graph neural networks. Furthermore, we propose a relational graph convolutional model that suits this representation. Our experimental study demonstrates this model's efficiency in performing video-related tasks while achieving comparable performance to the current state-of-the-art. An interesting avenue for future work is to explore new graph representations of videos, including learnable methods. Additionally, we consider the development of new dedicated graph neural models for processing the unique and dynamic structure of the video-graph as an interesting research direction. Finally, unified models for image and video understanding that disregard temporal edges could be explored in order to take advantage of the amount of data in both worlds.

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
