# OpenReview forum: "GraphVid: It Only Takes a Few Nodes to Understand a Video"
_thecvf.com/ECCV/2022/Workshop/VIPriors — VIPriors 2022 OralPosterTBD_

### Official Review · Reviewer_Uiac · 2022-08-01
**The idea presented in the paper is simple but can effectively speed up action recognition, therefore the paper should be accepted.**

**Rating:** 6
**Confidence:** 5

**Review:**

Summary:

- The authors propose an efficient graph video representation, GraphVid, that can be used for action recognition with reduced time and memory requirements. GraphVid results in a large efficiency gain without decreasing the performance.


Positive points:

+ The idea presented in the paper is interesting and can facilitate future work on action recognition.
+ Paper experiments 4 augmentation strategies to improve the model performance.
+ Thorough experiments and ablation study show GraphVid effectiveness.
+ Competitive results on two action recognition benchmarks, Kinetics-400 and Charades.


Negative points:

- Relevant related work is missing. GCN have been used for video modelling before.
1) Yan, Sijie, Yuanjun Xiong, and Dahua Lin. "Spatial temporal graph convolutional networks for skeleton-based action recognition." Thirty-second AAAI conference on artificial intelligence. 2018.
2) Thakkar, Kalpit, and P. J. Narayanan. "Part-based graph convolutional network for action recognition." arXiv preprint arXiv:1809.04983 (2018).
3) Korban, Matthew, and Xin Li. "Ddgcn: A dynamic directed graph convolutional network for action recognition." European Conference on Computer Vision. Springer, Cham, 2020.
4) Papadopoulos, Konstantinos, et al. "Vertex feature encoding and hierarchical temporal modeling in a spatial-temporal graph convolutional network for action recognition." arXiv preprint arXiv:1912.09745 (2019).
...

- Writing is sloppy and overly complex in places. The text can be simplified by removing sentences such as "to be defined hereunder" (line 207), "The following is a description of how we utilize the superpixels to construct our video-graph representation." (line 202-203)...

- Spatial edges and figure 5. It is unclear whether the spatial graph for each frame is complete. I do not see an explanation about edge selection, however, it seems that in figure 3 only "neighbouring super pixels" are connected.

- Missing algorithm time complexity for graph generation (i.e. extraction of super pixels, graph construction).

- Prior knowledge incorporation (line 369): I do not see how optical flow is currently encoded in the graph video representation, especially due to the absence of coordinates. For example, if an object moves fast within consecutive frames, the distance between the respective super pixels over time might be larger than d_proximity. This way, information about the object motion (direction) is lost completely.


Justification:
The idea of using super pixels in combination with GCNs is, to my knowledge, novel. The experiments are thorough and show the effectiveness of the method. The paper needs some fixes in the text as indicated above. My rate is weak accept.

---

### Official Review · Reviewer_1QBf · 2022-08-04
**Nice new idea to process video with GCNs**

**Rating:** 10
**Confidence:** 5

**Review:**

I can only recommend to accept this paper. It is highly related to the workshop due to many reasons.
1) Authors propose GraphVid. A new idea based on GCN to process video in an efficient way.
2) Apart from being a new and nice idea, it can offers state-of-the-art performance.
3) GraphVid uses not only opt-flow as prior knowledge but also many new specific data augmentation methods to extract information from the data in the most efficient way possible.
4) GraphVid allows to make a better use of the data and reduce the computational burden.
5) All ideas are support by experiments.
6) The paper is very easy to read.
... Many others

---

### Decision · Program_Chairs · 2022-08-08

**Decision:**

Accept (Oral/Poster TBD)

**Comment:**

Dear authors,


Congratulations! Your work has been accepted to the VIPriors workshop. Decisions on oral/poster presentations will follow later, when the program of the workshop is finalized.

*Please note the first action item is due on Wednesday! Please see instructions below.*

**Camera-ready instructions**

There is some work left to be done to ensure your work is included in the ECCV conference workshop proceedings. The ECCV publication managers use CMT to collect all workshop papers. This means we will migrate your paper from the VIPriors OpenReview page to the centralized ECCV workshop proceedings CMT page. The VIPriors program committee will ensure the details of your work (name, title, email address) are transferred to the CMT page, after which the ECCV proceeding managers will invite you to upload the camera-ready version of your work to the centralized ECCV CMT workshop proceedings page.

Please carefully follow the following instructions:
- **Before August 10th**, ensure that the first author has a CMT account under the same email address as the OpenReview account through which the accepted work was submitted. This account will be used to invite you to upload the camera-ready paper.
- Fill out this form, to inform us that the CMT account is in order: https://docs.google.com/forms/d/e/1FAIpQLSfyAoPv2_srESKaLRHIsHoWe3Fss1Z50ykdH7SzZpenA0m_5g/viewform
- Await instructions from the ECCV publication organizers, sent through CMT, on how to submit your camera-ready paper.
- Submit the camera-ready paper **before August 22nd**. Follow the camera-ready instructions for the main conference: https://eccv2022.ecva.net/submission/call-for-papers/.

**Attending the workshop**

We invite all authors of accepted works to attend the workshop in person on October 24th 2022 at ECCV in Tel Aviv. Please note a conference registration is required to attend the workshop. The workshop will be hybrid, enabling both in-person and remote attendance. We hope all accepted works can be represented in-person by at least one author, but we understand if this is not possible. Remote attendance of the workshop will be possible, though unfortunately there are limits on presenting works remotely: we intend to enable remote oral presentations, but this is not possible for posters.

Please fill out this form *before September 26th* to inform us of your attendance: https://docs.google.com/forms/d/e/1FAIpQLSfqRhdd2pq8t4CC8hL_c8fQo_TWcbzuQH3KGLzKVE36iTW_oQ/viewform.

**Presenting your work at the workshop**

Authors of all accepted papers are invited to present a poster at the workshop. Instructions on poster format will follow at a later date, but we will ask you to print and bring your own poster to the workshop.


For more information, as well as updates on the program of the workshop, keep an eye on our website: https://vipriors.github.io.

We thank you for choosing to submit to our workshop, and we are very much looking forward to hosting you in person in Tel Aviv!


Kind regards,

Robert-Jan Bruintjes
VIPriors program committee